# HiRT: Enhancing Robotic Control with Hierarchical Robot Transformers

**Jianke Zhang**[* 1], **Yanjiang Guo**[* 1], **Xiaoyu Chen**[1],
**Yen-Jen Wang**[2], **Yucheng Hu**[1], **Chengming Shi**[1], **Jianyu Chen**[† 1,3]
[1]Institute for Interdisciplinary Information Sciences, Tsinghua University
[2]University of California, Berkeley
[3]Shanghai Qizhi Institute
{zhangjk24, guoyj22, chen-xy21, huyc24, shicm19}@mails.tsinghua.edu.cn,
wangyenjen@berkeley.edu, jianyuchen@tsinghua.edu.cn

**Abstract:** Large Vision-Language-Action (VLA) models, leveraging powerful pre-trained Vision-Language Models (VLMs) backends, have shown promise in robotic control due to their impressive generalization ability. However, the success comes at a cost. Their reliance on VLM backends with billions of parameters leads to high computational costs and inference latency, limiting the testing scenarios to mainly quasi-static tasks and hindering performance in dynamic tasks requiring rapid interactions. To address these limitations, this paper proposes **HiRT**, a **Hi**erarchical **R**obot **T**ransformer framework that enables flexible frequency and performance trade-off. HiRT keeps VLMs running at low frequencies to capture temporarily invariant features while enabling real-time interaction through a high-frequency vision-based policy guided by the slowly updated features. Experiment results in both simulation and real-world settings demonstrate significant improvements over baseline methods. Empirically, in static tasks, we double the control frequency and achieve comparable success rates. Additionally, on novel real-world dynamic manipulation tasks which are challenging for previous VLA models, HiRT improves the success rate from 48% to 75%.

**Keywords:** Imitation Learning, Robots, Vision Language Models

## 1 Introduction

Large Vision-Language-Action (VLA) models [1, 2] provide a principled way to combine large vision-language models (VLMs) [3, 4, 5, 6] with end-to-end training on embodied tasks. Building on the top of pre-trained VLMs, existing VLA models [1, 2] propose to tune VLMs on massive robot data, which enables the direct end-to-end robot control while enjoying the benefits of VLM pretraining. Existing works mostly focus on multi-task generalization, enhancing performance in zero-shot and few-shot learning across various tasks.

Though the VLM backends with billions of parameters bring superior generalization advantages, it comes at the cost of the heavy computational burden. During deployment, it results in low control inference speed and high latency. This can slow robot movements and extend task completion times, impairing performance and safety in dynamic tasks like manipulating fast-moving objects in cluttered environments [7, 8]. The control frequency limitations of large VLA models remain a significant obstacle to deploying these advanced models on real-world robots.

Inspired by the dual process theory of human cognition [9], we propose HiRT, a hierarchical interactive imitation learning framework for VLA models. Dual process theory posits that there are two systems

---

[*]These authors contributed equally to this work.
[†]Corresponding authors.

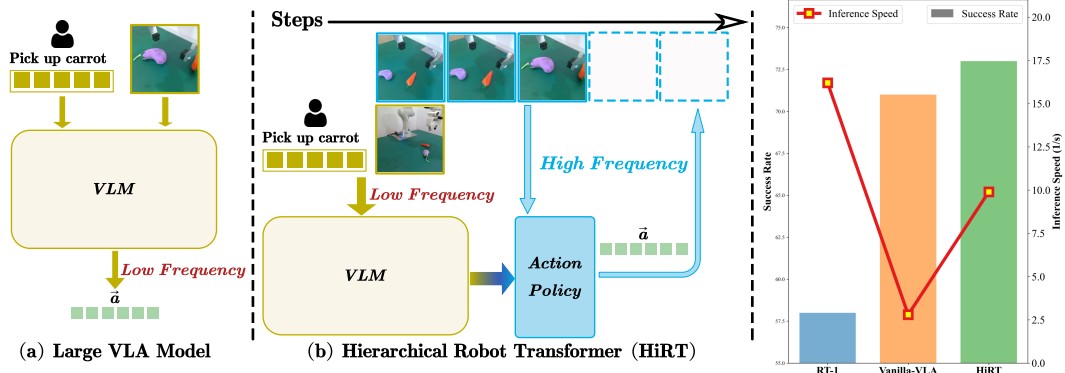

**Figure 1: Illustration of our proposed HiRT high-level architecture. (a)** Unlike large VLA models that directly output low-level actions with VLM, **(b)** HiRT is a hierarchical policy based on VLM. Given a task language instruction, the VLM encodes the observations into features that integrate multimodal information, and then a lightweight action policy conditions this latent to generate low-level actions asynchronously. As shown in **(c)**, our method can achieve higher performance and significantly improve inference speed.

in human cognition: System 1, responsible for fast, intuitive reactions, and System 2, responsible for slow, analytical planning. Current VLA models can be seen as relying solely on System 2, using computationally expensive VLMs for inference and action generation. However, we argue that VLA models can benefit from merging these systems. HiRT utilizes System 2 to extract high-level, slowly changing information that guides a lightweight System 1 module. This System 1, implemented by a smaller model, can react swiftly to environmental changes. Though lightweight, System 1 in HiRT can leverage the guidance from System 2 to maintain performance comparable to the original VLM while obtaining notable speed gain.

We term this approach HiRT, a hierarchical interactive imitation learning framework designed for rapid execution across a variety of instructions, scenes, and tasks. HiRT consists of two primary components: the understanding module and the execution module. The understanding module, InstructBLIP (7B) [5], is a pre-trained large visual-language model that transforms visual information and language instructions into latent features enriched with commonsense knowledge for long-term scene understanding, including task planning and error correction. The execution module is a compact visual-based action policy that processes short-term scene cognition, utilizing prior observations and latent features from the visual-language model. To enhance focus on global instruction and visual data, we incorporate novel conditioning layers within the execution module. HiRT leverages the slower visual-language model to guide the swift low-level policy, enabling efficient performance in both quasi-static and dynamic tasks at high frequencies. Additionally, we achieve further speed optimizations by adjusting the asynchronous frequency of the modules.

## 2 Related Works

**Language-Conditioned Imitation Learning for Robot Manipulation.** The study of integrating language with robotic actions [10, 11, 12] through imitation learning has a long history, where language is commonly used as goal specification [13, 14, 15, 16] or intermediate representation for planning [17, 18, 19]. Some prior works have employed reinforcement learning techniques [20, 21, 22, 23, 24] to solve certain types of downstream tasks. To address incapability in generalization of these RL methods, recent works concentrate on prompting Large Language Models (LLMs) [17, 25, 26, 27, 28] for high-level task planning and fine-tuning vision-language models (VLMs) on expert robotic datasets for low-level robotic control [20, 13, 24, 29, 30, 31]. Different from previous works that explore how to generalize to new tasks, we focus on solving low-level manipulation tasks by leveraging the extensive visual-linguistic knowledge within VLMs more efficiently and effectively.

**Vision-Language Models for Robotics.** Applying pre-trained VLMs [3, 4, 5, 6, 32] to various embodied scenarios is a recent focal area of research. Most of the prior works focus on using VLMs for high-level planning or reasoning [27, 33, 34, 35, 36, 37, 38]. To effectively connect visual or linguistic information with the physical environment, embodied models need to fine-tune pre-trained VLMs on embodied data [1] including video data containing task-level planning in linguistic form [39, 17, 27], simple text descriptions [40, 41], low-level actions [42, 43, 44] (known as vision-language-action models). However, deploying such large VLA models often results in slow inference speeds [45], which makes embodied models unsuitable for scenarios requiring precise operations or quick execution. Our approach focuses on addressing this limitation by using a novel policy model, which can effectively retain the robust visual-linguistic capabilities of the larger models.

**Hierarchical Action Planning.** Hierarchical action planning [17, 46, 27, 47, 48] involves decomposing a task into multiple simpler tasks that can be executed directly, enabling strategies to tackle more complex, long-horizon tasks. Previous works have demonstrated the role of inputting prompts into LLMs as a bridge to low-level actions. Specifically, this can be implemented through task-level planning [49, 39, 46], code execution [50, 51, 52], or other planning representations such as 3D scene graph [53], affordance function [54], and action pattern for locomotion [55]. However, these approaches are typically agnostic to physical embodiment, preventing the high-level models from directly interacting with the physical environment. In contrast to these methods, we ground VLMs to a specific robot's physical form in an end-to-end manner, enabling it to learn hierarchical task planning through continuous intermediate representations.

## 3 Method

In this section, we first establish the problem in Sec.3.1. Then, we present HiRT, a hierarchical policy architecture that supports multi-task learning and fast inference in Sec.3.2. The key intuition is to draw help from pre-trained VLMs to extract rich semantic representations from multi-modal inputs, and then apply these representations to lightweight action policies that can operate asynchronously and independently of the VLM. Specifically, HiRT explores a popular vision-language model, InstructBLIP [5], utilizing its open-source model as the backbone. We aim to output low-level actions with a latent-conditioned policy that leverages historical observations and latent encoded by VLM. This small-scale policy should operate independently of the large model at a higher frequency, necessitating a compact architecture composed of lightweight visual encoder. Following BC-Z [15] and RT-1 [13], we design a latent-conditioned model as the lower-level policy, capable of independently performing behavior cloning for a limited number of tasks at high frequency.

### 3.1 Problem Formulation and Method Overview

The language-conditioned manipulation problem can be considered a decision sequence under the environment modeled by Markov decision process: $(S, A, R, P, \rho_0)$, where $S, A, \rho_0$ represents state space, action space and initial state distribution respectively, $R : S \times A \times S \to \mathbb{R}$ represents the reward function, indicating whether a wanted state or task has been completed, $P : S \times A \times S \to [0, 1]$ represents probabilistic forward dynamics function of the environment. Specifically, given a free-form language instruction $l$ specifying a certain task, the control policy receives a visual observation $\boldsymbol{o}$ which is typically composed of a series of images. Then an action $\boldsymbol{a} \in A$, incorporating the relative position and pose of the end effector, is sampled from an action distribution $\pi(\cdot|\boldsymbol{o}, l)$ modeled by control policy.

For HiRT, the policy $\pi(\boldsymbol{a}|\boldsymbol{o}, l)$ is parameterized by $F_\theta$ from the vision language model and $S_\phi$ from the swift latent-conditioned policy. At certain time steps in the trajectory $\hat{t}_k \in \{t_i\}_{i=1}^T, k \le T$, the VLM backbone takes in a visual observation $\bar{\boldsymbol{o}}_{\hat{t}_k} = Sample(\boldsymbol{o}_{:\hat{t}_k})$ obtained through asynchronous sampling and a natural language instruction $l$, and outputs a fused embedding: $\boldsymbol{z}_{\hat{t}_k} = F_\theta(\bar{\boldsymbol{o}}_{\hat{t}_k}, l)$. Simultaneously, at each time of step, the latent-conditioned model predicts actions with recent context of visual observations and the latest latent: $\boldsymbol{a_t} = S_\phi(\boldsymbol{o}_{:t}, \boldsymbol{z}_{\hat{t}_k})$. The specific details of the modules will be explained in the following section.

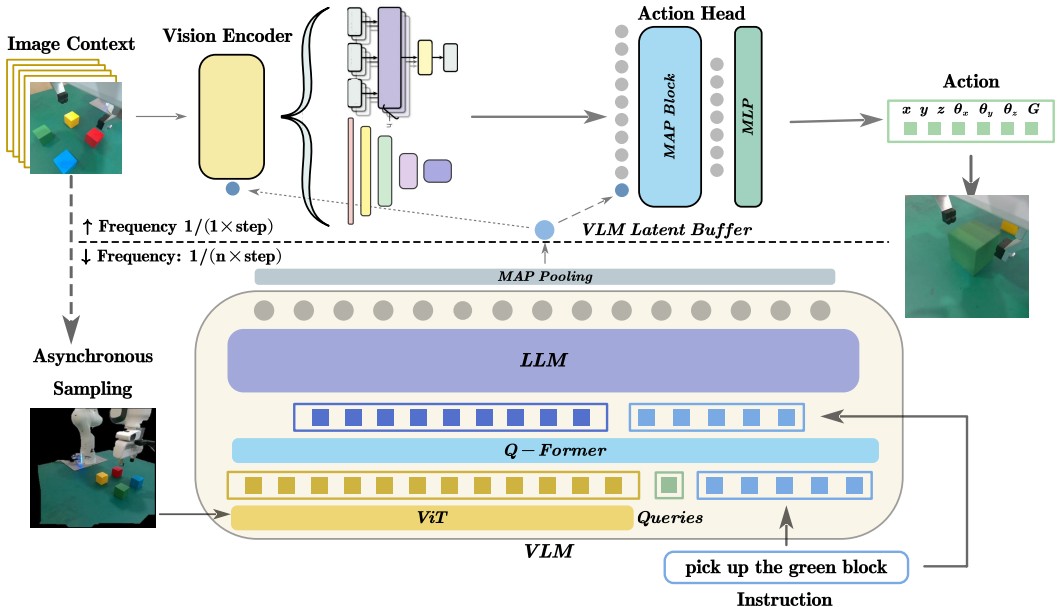

**Figure 2: HiRT network architecture.** The instruction is transformed into a continuous latent with sampled visual observation with a vision-language model and is cached into a latent buffer. At each step of inference, the pre-trained vision encoder encodes visual observations conditioned on the latest latent, and then the reduced vision-language tokens are decoded to low-level action with a conditioned action head.

## 3.2 The HiRT Framework

### 3.2.1 Encoding Multi-modal Information with Vision-Language Model

In HiRT, InstructBLIP [5] encodes the instruction $l$ using a visual signal $\bar{o}$ in the form of a single image. InstructBLIP comprises a pretrained visual encoder, a large language model (LLM), learnable query tokens, and a Q-Former [3]. At each execution time step $\hat{t}_k$, the visual observation (from either the wrist or third-view camera) is encoded by a Vision Transformer (ViT) [56] into a sequence of visual tokens:

$$\hat{X}^o_{t_k} = ViT(\bar{o}_{t_k}) \in \mathbb{R}^{N \times d}$$

where $N$ denotes the token length and $d$ the token width. Subsequently, $\hat{X}^o_{t_k}$ is concatenated with the instruction tokens $X^l_{t_k}$ and learnable query tokens $X^Q$, and encoded by the Q-Former (a lightweight transformer) into an image representation fused with semantic information:

$$X^o_{t_k} = QFormer(\hat{X}^o_{t_k}, X^l_{t_k}, X^Q)$$

Finally, these visual query features are used as prompts for the pre-trained LLM (LLaMA [57]). Set the embeddings at layer $i$ as $X^i_{t_k}$, the output at layer $i + 1$ is computed as follows:

$$\hat{X}^i_{t_k} = MSA(LN(X^i_{t_k})) + X^i_{t_k} \tag{1}$$

$$X^{i+1}_{t_k} = MLP(LN(\hat{X}^i_{t_k})) + \hat{X}^i_{t_k} \tag{2}$$

$$X^1_{t_k} = (X^o_{t_k}, X^l_{t_k}), X^i_{t_k} = (x^i_1, x^i_2, \cdots, x^i_N)_{t_k}, i = 1, \cdots, L \tag{3}$$

where $L$ denotes depth of transformer layers in the LLM, $MSA$ represents the multi-head attention module, $MLP$ stands for the multi-layer perceptron, and $LN$ denotes LayerNorm. Instead of generating language tokens from the final layer output $X^{L+1}_{t_k}$, we aim to use the informative language embeddings to guide action generation. We employ a MAP module [58], a single layer of attention block, to aggregate these representations: $\boldsymbol{x}_{t_k} = MAP(X^{L+1}_{t_k})$, which will be used for conditioning the action policy in Sec.3.2.2.

### 3.2.2 Latent-Conditioned Policy

Following the BC-Z [15] and RT-1 [13], which uses instructions and video as task embeddings, we encode the image context $o_{:t}$ into visual tokens $X_{:t}^v$ using a lightweight visual encoder, i.e., EfficientNet [59] and Vision Transformer [4]. Then, we use a MAP block to aggregate all the tokens into the continuous action space. To further integrate the informative task embeddings encoded by VLM, we make use of the following conditioning strategies on either the visual encoder or action head:

**FiLM-Condition.** For visual encoder based on convolutional network (CNN), each hidden layers are conditioned on the VLM latent variable $x_{t_k}$. In EfficientNet, We use FiLM layers to compute the conditioned features: $\hat{H} = FiLM(H \mid x_{t_k}) = W_\gamma x_{t_k} \cdot H + W_\beta x_{t_k}$, where $H$ represents the hidden features, and $W_\gamma, W_\beta$ are the learnable parameters in the FiLM layer.

**Condition with Cross-Attention Layers.** In each self-attention layers of Transformer, we insert an additional cross-attention layer for conditioning: $\hat{H} = CrossAttn(H, W_h x_{t_k}) + H$, where $W_h$ represents a learnable parameter that projects $x_{t_k}$ to the space of hidden tokens $H$.

**Condition with Prefix Tuning.** To better enable VLM to regulate low-level actions, we utilize the VLM latent variable $x_{t_k}$ as a prefix prompt for the MAP block in the action head. Specifically, the actions are computed by $a_t = MLP(MAP([x_{t_k}, X_{:t}^v]))$.

## 3.3 Training and Inference Strategy

**Asynchronous Operation and Sampling.** During the inference phase, we can accelerate the model by adjusting the execution frequency of the VLM. Specifically, at the initial time step $t = 0$, the VLM encodes multi-modal information with visual contexts and stores it in a cache. In subsequent steps, the latent-conditioned policy use the most recent latent variable from the cache to quickly output actions while the VLM runs asynchronously in parallel with the latent-conditioned policy. This asynchronous mechanism allows the policy to operate at nearly the same speed as the latent-conditioned policy, avoiding delays due to the VLM's slower inference. However, the asynchronous operation may cause the policy to use latent variables that reflect scene and instruction information from several steps earlier, which is misaligned with signals used in training. Therefore, during training stage, HiRT randomly selects a step from the past observation contexts $o_{:t}$ and uses the corresponding third-view image as the VLM's visual input. This technique can enhance robustness of the policy to the time-inconsistant latent variable.

**Training Objective** During training, the VLM part is finetuned with LoRA [60] while rest of the network is fully finetuned. Concretely, we utilize maximum likelihood imitation learning objectives. The desired relative position $a^{pos}$ of the end-effector (or continuous joint action) is optimized via regression loss (e.g. MSE loss). The discrete status $a^{end}$ of the end-effector is optimized with binary cross-entropy loss:

$$\mathcal{L} = \sum_{\mathcal{B}} \frac{1}{|\mathcal{B}|}(||a^{pos} - \hat{a}^{pos}||_2^2 + BCE(a^{end}, \hat{a}^{end}))$$

where $\hat{a}^{pos}, \hat{a}^{end}$ denote the demonstration for relative position and status of the end-effector in a sampled mini-batch $\mathcal{B}$.

## 4 Experiments and Analysis

In this section, we conduct extensive experiments across three domains, including two simulated benchmarks Metaworld [61] and Franka-Kitchen [62], and a real-world panda manipulation environment to verify the effectiveness of our HiRT framework. We first introduce experiment setups in Sec.4.1. Then we present a quantitative analysis of the performance on quasi-static tasks, evaluating HiRT's capability to enhance inference speed while preserving generalization performance in Sec.4.2. Additionally, we test the performance in real-world dynamic tasks in Sec.4.3. Finally, we discuss design choices for implementing HiRT and perform ablation studies on key modules in Sec.4.4.

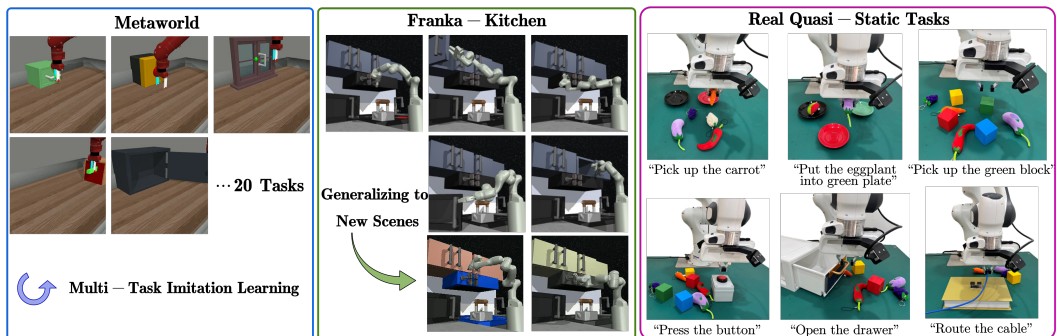

**Figure 3:** Visualization of the tasks in three domains. The left is Metaworld [61] in which we focus on the ability to learn multi-tasks. The middle depicts Franka-Kitchen [62] in which we study the ability to generalize to new scenes. The right shows our real-world settings, in which the model is trained on simple quasi-static tasks and tested on much more complex scenarios with unseen objects.

| Method | | Metaworld | Franka-Kitchen | | Real-world | | | | |
|---|---|---|---|---|---|---|---|---|---|
| Setting | Infer | Seen | Seen | Unseen | Pick- | Button- | Cable- | Drawer- | Average |
| | Speed/Hz | 20 Tasks | 5 Tasks | 2 Tasks | Place | Press | Route | Open | |
| RT-1 | 20.1 | 65.8 | 63.8 | 33.0 | 35.0 | 70.0 | 55.0 | 40.0 | 55.0 |
| DP | 4.6 | 52.2 | - | - | 30.0 | 70.0 | 30.0 | 50.0 | 45.0 |
| Vanilla-VLA | 4.1 | 73.8 | 73.4 | 70.0 | 70.0 | 85.0 | 65.0 | 65.0 | 71.3 |
| HiRT | 9.8 | 76.4 | 80.8 | 76.0 | 70.0 | 90.0 | 60.0 | 60.0 | 70.0 |

**Table 1:** Success rates on quasi-static manipulation tasks.

## 4.1 Experiment Setup

**Simulation Setup.** The Metaworld benchmark provides 50 distinct tabletop manipulation tasks, in which we use 20 tasks (each with 50 expert demonstrations) for multi-task learning. Franka-Kitchen includes 5 kitchen manipulation tasks. Following Nair et al. [19], we train policy models on 100 expert demonstrations for each task and test on tasks in origin and two new scenarios (alter the color scheme of the scene). We record the success rate to assess task performance: 20 attempts for each task in Metaworld and 100 for each task in Franka-Kitchen. To evaluate inference speed, we directly measure average time the policy takes to process 100 frames (avoiding influence of rendering).

**Real World Setup.** Our real-world experiments involve multiple quasi-static manipulation tasks on the Franka Emika Panda robot, involving picking and placing various objects, routing cables, pressing buttons, and opening drawers. Specifically, we collect 2000 trajectories including image observations from wrist and third-view cameras. For quasi-static tests, we place many other objects on the table to introduce distractions and we also test whether the model can grasp entirely new objects it has never seen before to verify its semantic grounding capabilities. Besides, we test the policy's performance on dynamic tasks by moving the target object at a roughly constant speed while the robotic arm executes its actions. All tasks involve randomization (e.g. the object's position, type, number of distracting objects, and the initial state of the gripper). We report success rate of each task over 20 attempts and the average time cost during real-world roll-out. More details on the design of experimental scenarios can be found in Appendix A.2, which can better demonstrate our testing of the generalization ability of semantic grounding in real scenarios.

## 4.2 Performance on static manipulation tasks

We train HiRT on 20 tasks from Metaworld, 5 tasks from Franka-Kitchen, and 4 skills from the real world (as shown in Figure 3). For comparison, we evaluate Diffusion Policy (DP) [63], Vanilla-VLA, which directly outputs actions from VLM (reimplementation of RT-2 [1]), and RT-1 [1] method under the same settings.

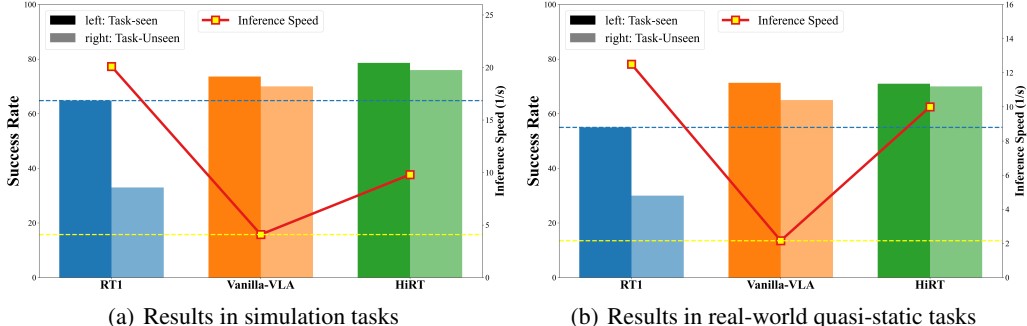

| (a) Results in simulation tasks | (b) Results in real-world quasi-static tasks |

**Figure 4:** Speed and Performance of HiRT with different VLM frequency. Each chart compares the different states of HiRT with Vanilla-VLA and RT-1.

**Imitation and Zero-Shot Generalization Performance.** Table 1 presents the experimental results in simulation and real-world environments. HiRT achieves the highest success rates in simulated tasks and a high level of generalization capability similar to Vanilla-VLA in real-world environments. Compared to RT-1, which uses language embeddings for conditioning, HiRT, which utilizes vlm latent for conditioning, shows an average 20% higher success rate on seen tasks and a 30% higher success rate on new task scenarios. This demonstrates that VLM can leverage visual scenes to provide better instruction embeddings, aiding the small action policy in generalizing to new tasks.

**Balancing between Performance and Efficiency.** To further demonstrate HiRT's ability to balance performance and execution frequency, we evaluate the model's inference speed and task success rate. Results are shown in Figure 4. Notably, HiRT's performance is comparable with Vanilla-VLA, while its inference speed significantly increases to 9.8Hz, nearly doubling the original speed. This indicates that HiRT can significantly enhance inference speed while maintaining model generalization capabilities.

## 4.3 Performance on real-world dynamic manipulation tasks

We evaluate models with varying frequencies and performance levels on a series of dynamic tasks in real-world scenarios. Specifically, as illustrated in Figure 5, we simulate realistic operational tasks by moving target objects at a constant speed of 1 cm/s, presenting different levels of generalization.

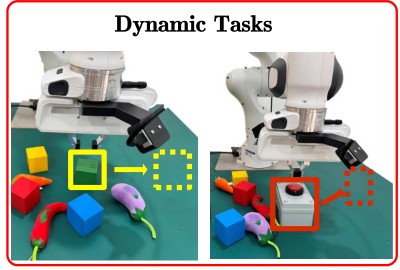

**Figure 5:** Visualized Dynamic Tasks.

| Method | Time/s↓ | Seen↑ | Unseen↑ | Average↑ |
|---|---|---|---|---|
| Diffusion Policy | 10.38 | 20.0 | 15.0 | 18.0 |
| RT-1* | 14.22 | 25.0 | 10.0 | 18.0 |
| Vanilla-VLA | 9.25 | 55.0 | 40.0 | 48.0 |
| **HiRT** | **6.18** | **80.0** | **70.0** | **75.0** |

**Table 2:** Success rates on real-world dynamic manipulation tasks. With our hierarchical design, HiRT achieves the highest success rate and finishes the task in the least time.

Results are shown in Table 2, where the column of *Time* represents the duration taken by the model to complete the quasi-static task in the same scene without moving objects, serving as a reference of the model's efficiency. HiRT achieves the shortest completion time in quasi-static tasks and the highest task success rate in dynamic action tests within both in-domain and out-domain scenarios, indicating its strong generalization capability while maintaining a high execution speed. Overall, the HiRT approach effectively applies VLM-based methods to various dynamic tasks. A comparision example is visualized in Figure 6.

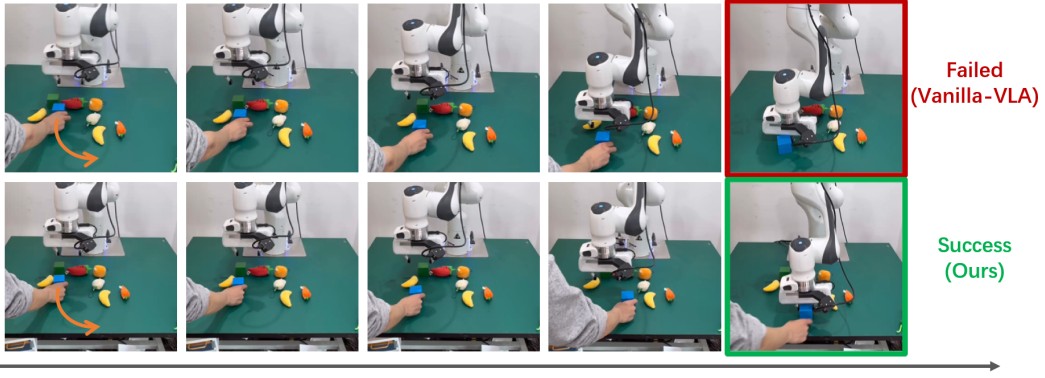

**Figure 6:** Comparisons under dynamics real-world experiments. The blue block is moving along a trajectory on the table. Our method successfully tracks the movement of the blue block and catches it. While the baseline method misses the blue block due to long inference time and high latency.

## 4.4 Ablation Study on implementation of HiRT

In this experiment, we seek to understand the different components of HiRT. Specifically, we compare the full HiRT with **HiRT(-IC)**, which uses the most recent single frame for latent-conditioned-policy, **HiRT(-CD)**, which replaces the conditioned-transformer and conditioned MAP (keep FiLM block which has been validated as effective in RT-1 [13]) with the original module. We primarily conduct module ablation in the Franka-Kitchen environment because it allows for rapid testing on generalization capability. More ablations can be found in Appendix A.3.

| Method | Infer Speed↑ | Knob | Ldoor | Light | Micro | Rdoor | $Avg_{seen}$ | Light* | Rdoor* | $Avg_{new}$ |
|---|---|---|---|---|---|---|---|---|---|---|
| RT-1 | 20.1 | **90** | **72** | 51 | 87 | 19 | 63.8 | 1 | 65 | 33 |
| Vanilla-VLA | 4.1 | 84 | 43 | 96 | 67 | 77 | 73.4 | **85** | 46 | 65.5 |
| HiRT-IC | 10.1 | 83 | 0 | 93 | 18 | **100** | 58.8 | 28 | 66 | 47 |
| HiRT-IC-CD | 11.0 | 87 | 35 | 39 | 6 | 73 | 48.0 | 0 | 19 | 9.5 |
| HiRT-CD | 10.9 | 79 | 14 | 74 | 64 | 90 | 64.2 | 39 | 84 | 61.5 |
| **HiRT** | 9.8 | 84 | 43 | **99** | **79** | 99 | _80.8_ | 52 | **100** | _76.0_ |

**Table 3:** Ablating Components of HiRT. Results with *-IC* reveal that image context is important for good performance. Using the combined conditioning strategy leads to a 20% increase in success rate.

**Does VLM-Latent Conditioning Improve Multi-Task Performance?** As shown in Table 3, the full HiRT model achieves the highest task completion rates in both multi-task and new task settings. Removing the extra conditioning layers (HiRT-CD) results in a 20% decrease in success rate, highlighting the importance of conditioning for multi-task learning.

**How Does the Latent-Conditioned Policy Perform with Different Visual Inputs?** In Table 3, regardless of the model structure (e.g. full, -CD), using multiple consecutive visual inputs significantly outperforms using a single image (results with *-IC*). Although using multiple visual contexts as input can reduce inference speed, it greatly enhances the policy's understanding of the scene, enabling it to generate more accurate actions using the same VLM-Latent output.

## 5 Conclusions, Limitations and Future Works

In conclusion, this study addresses the limitations of VLMs in handling complex dynamic tasks due to high computational costs and inference delays. By proposing HiRT, a hierarchical imitation learning framework, we enhance execution speed and multi-task generalization. However, due to data constraints, we have not yet employed HiRT on more complex dynamic tasks, such as grasping high-speed flying objects or rapidly adjusting object postures. This could be an area for our future research with more specific task data and adaptive adjustments to certain modules.

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

# A Appendix

## A.1 Implementation Details

During implementation, we take use of pretrained EfficientNet-B3 [59] and ViT-B/16 [56] for the vision encoder of low-level policy, which have been pretrained on large vision data. In training, we insert adapter layers (LoRA layers) throughout the entire InstructBLIP model, including the ViT, Qformer, and LLaMA. In the simulation results, the low-level policy utilize the former CNN architecture, while in the real-world results, the transformer based ViT architecture is employed. For simulation, the fast policy mainly contains the pretrained EfficientNet-B3 vision encoder and the FiLM layers, with totally about 35M parameters. For real world, the fast policy mainly contains the pretrained ViT-B/16 and the cross attention layers, with 150M parameters.

## A.2 More Details of Experiment Setup

### A.2.1 About Choice of Design for Test Scenarios

In the Metaworld simulation environment, we randomly initialize the positions of objects and the robotic arm during testing to assess the method's robustness to object positions in a multi-task setting. In the Franka-kitchen environment, we randomize the relative positions between the robotic arm and the operating platform for each test. Additionally, we significantly alter the color scheme of the scene to evaluate whether the method could complete specific tasks in a visual background that differs greatly from the training data. These two simulation environments are widely used in many studies to validate the fundamental generalization capabilities of models, particularly their robustness to changes in scenes and object positions.

In our real-world scenarios, the training data only includes situations with a few objects placed on a tabletop. During testing, we not only randomize the positions of the objects and the robotic arm but also place many other objects on the table to introduce distractions. Furthermore, we also test whether the model can grasp entirely new objects it has never seen before to verify its semantic grounding capabilities. Additionally, it's worth mentioning that HiRT's input does not include state information, so all generalization capabilities come from visual and language information.

### A.2.2 About evaluation of generalization capability

Our tests primarily focus on real-world experiments to validate the semantic generalization capability of our method. In the Metaworld and Franka-kitchen simulation environments, we mainly evaluate the method's generalization to different positions and visual scenes. In the real-world setting, we test whether the model can complete tasks despite the introduction of more distracting objects, a broader range of positional variations, diverse backgrounds, and entirely new objects, e.g. different shapes of vegetables, an arrow-shaped paper, unseen vegetables, toy pizza and blocks with unseen color.

### A.2.3 About data collection details

For the simulation environment data, we follow the setups of Metaworld and Franka-kitchen by using scripted policies to collect action trajectories for different tasks. In the real world, our data collection is carried out using both manual and scripted methods.

Specifically, for tasks such as grasping two types of toy fruits (carrot and eggplant), opening drawers, and routing cable tasks, we collect demonstrations manually using a remote operation joystick, ensuring that the target objects are roughly evenly distributed in the field of view. For grasping blocks of different colors, we use scripted policies. We fix four placement positions for the blocks and randomly initialize the robotic arm's position to collect trajectories for these tasks. ( Although the block positions are fixed, we find that HiRT could also correctly grasp blocks placed in novel locations not encountered during training.)

### A.3 More Ablation

**How Does Random Sampling in VLM Visual Inputs Affect Model Performance?**

To determine the impact of sampling images for VLM inputs, we compared the performance of HiRT with HiRT-AS under VLM settings of interval 1 and interval 6. As shown in Table 4, when VLM synchronizes with the action policy at every step during testing, HiRT-AS slightly outperforms HiRT. This is expected since HiRT-AS uses the most recently updated latent at each step during training. However, when VLM operates with an interval of 6 steps for inference, HiRT-AS shows nearly a 10% decrease in success rate compared to the original method, indicating that random sampling helps HiRT maintain better generalization performance during asynchronous operation.

|         | VLM-step | Frequency↑ | $Avg_{seen}$ | $Avg_{new}$ |
|---------|----------|------------|--------------|-------------|
| HiRT-AS | 1        | 1.97Hz     | 96.7         | 77.2        |
| HiRT    | 1        | 1.97Hz     | 94.2         | 76.0        |
| HiRT-AS | 6        | 13.42Hz    | 52.2         | 50.0        |
| HiRT    | 6        | 13.42Hz    | 63.4         | 52.5        |

**Table 4:** Importance of Asynchronous Sampling.

