# OpenReview forum: "HiRT: Enhancing Robotic Control with Hierarchical  Robot Transformers"
_robot-learning.org/CoRL/2024/Conference — CoRL 2024_

### Official Review · Reviewer_4RK9 · 2024-07-15
**This well-structured paper presents a compelling experimental section and convincing results, demonstrating notable advancements in model architecture.**

**Originality:** 4
**Technical Quality:** 4
**Clarity Of Presentation:** 4
**Potential Impact:** 3
**Recommendation:** 3
**Confidence:** 5

**Review:**

**Quality and clarity**

The paper is very well-written, striking a balance between clarity and providing sufficient technical detail. The authors' ability to replicate and train nuanced models like RT-1 and RT-2, which demand significant computational resources, is particularly impressive. The Related Works section is also commendable. Some minor details are missing, which I will inquire about in the rebuttal.

**Originality and significance**

Building upon RT-2, a VLM model fine-tuned for robotics actions, the paper proposes a seemingly obvious yet unexplored idea: splitting the model into "slow" and "fast" components. The community may have hesitated to implement this due to the perception that faster inference speed isn't a particularly attractive metric. However, the authors convincingly demonstrate that this approach is beneficial for dynamic tasks where rapid replanning is crucial for success. The insights, proposed architecture, and novelty of this work are valuable contributions to the field.

**Strengths**

The paper presents a well-chosen hypothesis with excellent execution. The experimental section is thorough, and the ablation studies are sufficient. The authors establish strong baselines and comparisons to well-known models (RT-1, RT-2).

**Weaknesses**

Given the formulated problem (best use of a VLM for robotics actions), there are no major weaknesses. Initially, my primary concern was the asynchronous nature of the VLM's interaction with the policy, potentially leading to stale latent representations. The authors address this with Asynchronous Sampling during training, explicitly accounting for this scenario. However, this crucial architectural choice was not ablated, which I recommend investigating.

**Quality Of The Limitations Section:**

3

**Questions For Rebuttal:**

Hello, thank you for your paper.

I have a few questions and suggestions to further enhance the paper:

1. Performance Comparison: It would be beneficial to include a performance comparison with the diffusion policy, even if it outperforms HiRT. This would provide valuable insights to the research community.
2. Appendix: Do you intend to provide the appendix mentioned in the paper?
3. Video Demonstration: A video showcasing dynamic tasks and a head-to-head comparison of inference speeds would significantly enhance the paper's impact.
4. Data Collection Details: The paper could benefit from more information about the collected data, including the data collection protocol and method (scripted policy or teleoperation).
5. Fast Policy Details: Please provide additional information about the fast policy, such as the number of parameters used.
6. Section 3.3 Clarification: The first paragraph of Section 3.3 could be rephrased to clarify the mechanics of the asynchronous interaction between the VLM and the policy.
7. Asynchronous Sampling Ablation: The paper mentions Asynchronous Sampling in Section 6 to address the issue of stale latents during training. This architectural choice seems crucial for the success of dynamic tasks, yet it wasn't ablated. I recommend including an ablation study for this.
8. Dataset and Model Release: I encourage you to release the dataset and the model. Additionally, consider releasing the replicated RT-1 and RT-2 models.

**Robotics Focus:**

4

**Summary Of Paper:**

The paper tackles the issue of slow inference speeds in Vision-Language Models (VLMs) for robotic actions. Using the RT-2 paper as a baseline VLM for robotic actions, the authors assert their approach achieves comparable performance on quasi-static tasks. Their method excels in dynamic manipulation tasks where an object is moved adversarially by an operator, emphasizing the need for fast inference to enable quick replanning. HiRT significantly improves the success rate in these scenarios from 48% to 75%.  Their approach leverages an open-sourced InstructBLIP pretrained VLM to generate embeddings of language-conditioned observations at a low frequency. A separate model with fewer parameters is then conditioned on the latent representation coming from the VLM. This allows the slower VLM to infer the current global state less frequently, while the second model operates at a higher frequency, enabling effective action in dynamic tasks.

**Summary Of Recommendation:**

Overall, I appreciate the proposed paper, its quality, and the results presented. The concept of splitting large VLM models into slow and fast components, while seemingly obvious, has not been thoroughly explored before, to my knowledge. The authors certainly deserve credit for their contribution in this area.  My "Weak Accept" rating stems from the possibility that other reviewers might perceive this as a small, incremental step within the vast design space of VLMs for robotics. Admittedly, from that perspective, this work doesn't represent a significant departure from the ideas presented in RT-1 and RT-2.

---

### Official Review · Reviewer_vpoa · 2024-07-19
**Review of HiRT**

**Originality:** 3
**Technical Quality:** 4
**Clarity Of Presentation:** 4
**Potential Impact:** 3
**Recommendation:** 3
**Confidence:** 4

**Review:**

Overall, this paper is well presented and showcases a nice execution of the system1-system2 idea. I appreciate the clear presentation of the method, the experimental results and ablation studies. Specifically,

Pros
- Concise, unambiguous illustrations e.g. Fig.1, Fig.2
- Well presented Section 3 (with some clarification questions below)
- The asynchronous inference setup provides a tunable execution frequency for the VLM, which helps balancing speed and performance given different VLM size, hardware etc.
- Experiment setup is overall relevant and of reasonable difficulty

Cons
- I would appreciate more diverse unseen tasks in both sim and real. This will help provide better understandings on the gains in generalization by using the proposed approach
- One main appeal for the proposed method is the ability to execute dynamic tasks, for which fast execution speed is a necessity. However, the experiments presented in the paper are limited to tracking a moving target object. It would be good to include more realistic tasks

**Quality Of The Limitations Section:**

3

**Questions For Rebuttal:**

- 134: Which vision encoder is used in the experiments?
- 161-167: Clarification: Are the VLM and action model co-trained where for every training step, gradients are propagated from the action model to the VLM?
- Table 2: How do you explain the poor performance of RT-1 here?
- 222: IC: is the vision encoder completely removed?
- 223: Re: CD, when both conditioned transformer and conditioned MAP blocks are removed/replaced, how does the VLM still connect with the action model?

**Robotics Focus:**

4

**Summary Of Paper:**

This paper presents an implementation of the system1-system2 concept, where a large vision language model is used for high level reasoning and better generalization, while a small model is used for outputting actions. The key appeal to this approach is that one may have the best of both worlds by having better generalization through the pretraining of the large model, while still having fast inference by running the small action model at a fast rate. The method presented in this paper uses a pre-trained 7B InstructBLIP model which is connected to the action model asynchronously through latent embeddings. The authors achieved both generalization and speed in the selected tasks, presenting a convincing case for the effectiveness of this approach.

**Summary Of Recommendation:**

Overall a well presented paper with strong experimental results. Some additional experimental results on generalization and dynamic tasks will help provide stronger arguments for the advantages of the approach.

---

### Official Review · Reviewer_FEy5 · 2024-07-22
**Review of HiRT: Enhancing Robotic Control with Hierarchical Robot Transformers**

**Originality:** 2
**Technical Quality:** 2
**Clarity Of Presentation:** 3
**Potential Impact:** 3
**Recommendation:** 2
**Confidence:** 5

**Review:**

**Pros:**

The paper aims to address a challenging problem in trying to use large VLM models for manipulation tasks with fast inference. Overall, the paper is well written and easy to follow.

**Cons / Questions:**

The overall approach seems reasonable. However, some of the design decisions are not fully ablated and it is unclear how significant these are.

First, the overall VLM approach to use InstructBLIP’s vision encoder followed with Q-former to get semantic vision-language representations makes sense. However, the query tokens from this model are then further sent through LLAMA (again together with the instruction tokens) to get the final instruction (or context) embedding. Since, the output of Q-former already has some vision-language grounding are all of the LLAMA layers important? or was it empirically found that Q-former’s grounding is not particularly good. The paper also mentions that LoRA was used for updating this VLM model. Can the authors add details on where were the adapter layers inserted (e.g .just for LLAMA or for InstructBLIP as well)?

This approach is somewhat broadly similar to RT-2 (which is the vanilla-VLA baseline according to the paper), wherein images are first processed by a VIT and then processed by a large LLM (together with the instruction tokens). However, importantly, in RT-2 the outputs of the LLM are the actual robot actions (tokenized). By contrast, in the current paper, the output of the LLM (LLAMA) is just the context for the low-level policy. This to me seems like a very important distinction. This is because, since RT-2 uses the LLM as the policy (and co-finetunes) it gets all the robustness and grounding from the VLM. However, in the current paper, the low-level policy is still trained on small robotics datasets and the LLM output is only used as context. Further, it is unclear if the low-level policy’s image encoders are pretrained (maybe the authors can comment on this). But in any case the low-level policy cannot really generalize or have very good semantic grounding since it hasn’t been exposed to large amounts of pre-training data. Further, the policy is being finetuned (without co-finetuning on VLM data) which means that its robustness will be lost.

Experiments: The above point is also reflected in the experimental section. The paper performs experiments on Metaworld and Franka-Kitchen. Neither of these environment suites test for any semantic generalization. However, the paper does mention that “test on tasks in origin and two new scenarios.” (Line 180). However, no other details are provided. I think this raises many questions —  How were these new scenarios created, what do they test? Is the vision model for low-level policy pre-trained? How do we get semantic generalization when we train our low-level policy on such few tasks.

Figure-3 in the paper shows some color variations for the franka-kitchen tasks. However, these variations seem minor. Another big problem with the Franka-Kitchen task is that the object positions/poses/geometry is fixed which means that even if semantics are changed the policy completely ignore and just use propriceptive input to complete the task. It is unclear, how any of these generalization questions were evaluated in these settings.

Dynamic real-world tasks:  The paper seems to mention that the policy is robust to moving the target object at a constant 1cm/s speed. How was this accomplished? It seems from some of the figures that this was done manually (please correct me if I am wrong). If this is indeed true, do you not see issues with the hand making the observations out-of-distribution (or was  data collected in this setting). Additionally, I wouldn’t really call a task where there is a slow-moving target a “dynamic task” since the dynamics of the task haven’t really changed because of motion. The task may require slightly more faster inference but clearly the dynamics of the task are still mostly the same.

Related work: I think the paper misses some of the related works, especially, which are concerned with dynamic tasks. For example, Saxena et al. look at a very similar problem of using VLMs for real-time control (i.e. fast inference). Similarly, there are other works which focus on using MPC for real-time control (although these works do not focus on VLMs), but I think they should still be cited and provided in context (since there is a huge body of work in enabling real-time control in robotics).

Saxena et al, MResT: Multi-Resolution Sensing for Real-Time Control with Vision-Language Models

Abeyruwan et al, Agile Catching with Whole-Body MPC and Blackbox Policy Learning

**Quality Of The Limitations Section:**

2

**Questions For Rebuttal:**

see above

**Robotics Focus:**

4

**Summary Of Paper:**

This paper focuses on the problem of enabling pretrained vision-language models (VLMs) to be used for robotic tasks with a high frequency. The overall goal is to be able to use pretrained VLMs (with potentially billions of parameters) to accomplish fast action inference which can in turn allow the robot to perform dynamic manipulation tasks. The paper accomplishes this using a heirarchical approach with a high-level VLM which uses InstructBlip (+  LLAMA) to provide instruction context. Similarly, a small low-level vision (and proprioceptive) uses this context to output robot actions at a high-control frequency. Experiments are performed on the meta-world and franka-kitchen tasks in simulation and some real-world experiments.

**Summary Of Recommendation:**

The paper aims to address a very important problem in robotics specifically, how can we get generalization from VLM while having fast inference. It is really unclear to me how generalizable the policies trained in the paper are. More details around experiment design, evaluation are needed to precisely understand the strength and weaknesses of the proposed approach.

---

### Author Rebuttal · Authors · 2024-08-08

### **Overall Response**
We sincerely appreciate all reviewers’ and ACs’ time and efforts in reviewing our paper. We thank you all for the insightful and constructive suggestions, which helped further polish our paper. We would like to clarify our Motivation and contributions as follows:

Using VLM as an action policy faces inference latency challenges. In this work, we employ a hierarchical structure to accelerate VLM inference while maintaining its generalization capability.

Here is a summary of our updates:

- We provide a **new video demonstration**, showcasing our real-world tasks setup, which is important  for understanding our evaluation of generalization, especially the semantic grounding capability.
- Section2 (related works): As suggested by Reviewer FEy5, we add two related works which are concerned with dynamic tasks. Both of which focus on using VLM or MPC for real-time control.
- Section3: As suggested by Reviewer 4RK9, we rephrase to clarify the mechanics of the asynchronous interaction between the VLM and the policy.
- Section4: We add some detailed explanation of our experiment setup and our experiments on real-world unseen tasks, including generalizing to new position, more distracting objects, and unseen new objects to better support HiRT's generalization capability.
- Section4: As suggested by Reviewer vpoa, we revise the description of the ablation experiments to make the design and purpose of these experiments clearer.
- Section5: The limitation section is specified in detail, and includes possible future works based on our method involving more complex dynamic tasks and real-world applications.
- Appendix: We add details that reviewers are concerned about, including adapter layers, model parameters, training details, data collection protocol, and ablation study of asynchronous sampling.

We hope we have addressed all your concerns and questions. Please let us know if there are still any concerns.

---

### Decision · Program_Chairs · 2024-09-04

**Decision:**

Accept

**Comment:**

Strengths:
+ HiRT tackles an important issue of slow inference speed in using VLMs for robotics.
+ The paper is well-written and easy to follow.
+ The figures are informative.
+ The experimental setup is clear and includes a good set of ablations.


Weaknesses:
- Evaluations do not extensively test for semantic generalization to highlight the generalization capability from VLM. The simulation experiments include only 2 unseen tasks on Franka-Kitchen. Franka-Kitchen also has limited pose-randomization of objects, which makes it easy to overfit.
- Evaluations are missing dynamic tasks that could highlight the benefit of fast inference speeds.
- Some details on adapter layers, data collection, and asynchronous sampling are missing.

Post rebuttal:
The authors provided an extensive rebuttal with additional experiments. One reviewer feels that some details are still missing, but the AC believes they can be added for the camera-ready.